# Safety and Efficacy of Weekly Paclitaxel and Cisplatin Chemotherapy for Ovarian Cancer Patients with Hypersensitivity to Carboplatin

**DOI:** 10.3390/cancers13040640

**Published:** 2021-02-05

**Authors:** Shinichi Tate, Kyoko Nishikimi, Ayumu Matsuoka, Satoyo Otsuka, Makio Shozu

**Affiliations:** Department of Gynecology, Chiba University Hospital, 1-8-1 Inohana, Chuo-ku, Chiba 260-8670, Japan; knishikimi@hospital.chiba-u.jp (K.N.); a-matsuoka@chiba-u.jp (A.M.); caxa5597@chiba-u.jp (S.O.); shozu@faculty.chiba-u.jp (M.S.)

**Keywords:** carboplatin, cisplatin, hypersensitivity reaction, ovarian cancer, platinum-sensitive disease, recurrence

## Abstract

**Simple Summary:**

This study was conducted to evaluate the safety and efficacy of weekly paclitaxel and cisplatin chemotherapy in patients with ovarian cancer who developed carboplatin hypersensitivity reaction. Eighty-six (86) patients who developed hypersensitivity reactions for carboplatin were treated with weekly paclitaxel and cisplatin chemotherapy, and 71 (83%) of the 86 patients were able to receive treatment without hypersensitivity reaction to cisplatin. The severity of the hypersensitivity reaction for cisplatin observed in all 15 patients was below grade 2, and there were no deaths due to hypersensitivity reaction to cisplatin. The majority of patients (55 patients, 64%) completed the scheduled weekly paclitaxel and cisplatin chemotherapy, and only 9 patients (10%) discontinued treatment due to hypersensitivity reaction within 6 cycles. Weekly paclitaxel and cisplatin chemotherapy were well-tolerated and effective for patients who developed carboplatin hypersensitivity reaction.

**Abstract:**

Background: This study aimed to evaluate the safety and efficacy of weekly paclitaxel and cisplatin chemotherapy (wTP) in patients with ovarian cancer who developed carboplatin hypersensitivity reaction (HSR). Methods: We retrospectively investigated 86 patients with ovarian, fallopian tube, and peritoneal carcinoma who developed carboplatin HSR during previous chemotherapy (carboplatin and paclitaxel) at our institution between 2011 and 2019. After premedication was administered, paclitaxel was administered over 1 h, followed by cisplatin over 1 h (paclitaxel 80 mg/m^2^; cisplatin 25 mg/m^2^; 1, 8, 15 day/4 weeks). We investigated the incidence of patients who successfully received wTP for at least one cycle, treatments compliance, progression-free survival (PFS), and overall survival (OS). Results: The median number of wTP administration cycles was 4 (Interquartile Range IQR, 3–7), 71 patients (83%) successfully received wTP, and 15 patients (17%) developed cisplatin HSR. The efficacy of treatment was as follows: 55 (64%) patients completed the scheduled wTP, 9 (10%) patients discontinued due to HSR to cisplatin within 6 cycles, 1 (1%) patient discontinued due to renal toxicity (grade 2) at the 6th cycle, and 21 (24%) patients discontinued due to progressive disease within 6 cycles. The median PFS and OS after administration of wTP were 10.9 months (95% CI: 7.7–17.7) and 25.9 months (95% CI: 19.0–50.2), respectively. Conclusions: wTP was safe and well-tolerated in patients who developed carboplatin HSR.

## 1. Introduction

The standard therapy for platinum-sensitive recurrent ovarian cancer is a platinum-based chemotherapy plus bevacizumab (Bev) [1] or poly (adenosine diphosphate [ADP]-ribose) polymerase (PARP) inhibitor maintenance [2] in case of response to platinum, and it is now influenced by the first-line treatment (use of Bev or PARPi). PARP inhibitors have been reported to be more effective in survival than Bev in patients with breast cancer susceptibility (BRCA) mutations in a network meta-analysis [3]. Moreover, a recent study has reported that carboplatin plus pegylated liposomal doxorubicin is the best chemotherapy regimen in combination with Bev for platinum-sensitive recurrent ovarian cancer [4]. Furthermore, the Desktop III trial revealed the impact of secondary cytoreduction in platinum-sensitive recurrent ovarian cancer [5], while the Gynecologic Oncology Group (GOG) 0213 trial did not [6].

However, one of the problems in the treatment of platinum-sensitive recurrent ovarian cancers is the occurrence of carboplatin hypersensitivity reaction (HSR) [7]. The frequency of carboplatin hypersensitivity reaction increases with repeated exposure [7]. After such a reaction has occurred, it is difficult to use platinum compounds, despite the patients having platinum-sensitive disease. The absence of platinum administration due to carboplatin hypersensitivity reaction consequently makes it difficult to determine the platinum sensitivity of each patient; therefore, they cannot benefit from poly (adenosine diphosphate [ADP]-ribose) polymerase (PARP) inhibitors [2], for which platinum sensitivity is a clinical biomarker [8].

Alternative treatments for patients who developed carboplatin hypersensitivity reactions include changing to non-platinum drugs, more intense premedication, changing to other platinum drugs [9], or using a desensitization therapy for carboplatin [10,11]. Desensitization therapy for carboplatin has been reported to be highly successful [10,11]; however, its administration is cumbersome and there are disadvantages in terms of temporal constraints. In addition, despite the use of an extensive desensitization regimen, two deaths have been reported [12]. Furthermore, administration of another platinum drug, nedaplatin [9,13], is not currently recommended as the frequency of hypersensitivity reaction to nedaplatin is relatively high (27%) [13]. In contrast, cisplatin administration for patients who developed carboplatin hypersensitivity reaction as alternative treatments have only been reported in a small number of patients, and most reports have mentioned only the short-term safety of cisplatin administration and not on its long-term efficacy [14,15].

Since 2011, we have performed weekly administration of paclitaxel and cisplatin (wTP) in consecutive patients who developed carboplatin hypersensitivity reaction. Compared to dose-dense paclitaxel plus triweekly carboplatin, wTP with low-dose cisplatin has been reported to produce less hematologic toxicity, renal dysfunction, and anorexia, although the therapeutic effects of both regimens were similar [16]. The aim of this observational study was to investigate the safety and efficacy of wTP in patients with carboplatin hypersensitivity reaction.

## 2. Materials and Methods

### 2.1. Patients and Study Design

This was a retrospective, single-center study. The study was approved by the Institutional Review Board (#3734) at Chiba University. We retrospectively enrolled a series of 86 consecutive patients who received wTP owing to carboplatin hypersensitivity reaction while undergoing paclitaxel carboplatin chemotherapy for ovarian, fallopian, and primary peritoneal cancer at our hospital from January 2011 to December 2019. Informed consent was obtained from all patients. Weekly paclitaxel and carboplatin were administered at initial treatment or at recurrent treatment to patients with platinum-sensitive disease during the study period. Patients who developed hypersensitivity reactions to carboplatin were administered wTP consecutively. Seventeen patients (20%) received carboplatin with slow infusion rates after hypersensitivity reaction to carboplatin; however, they again developed hypersensitivity. Desensitization therapy for carboplatin was not applied since the study was performed at an outpatient chemotherapy unit. The patients’ data included medical history, primary site, histology, stage, previous regimen, number of carboplatin cycles, severity of hypersensitivity reaction to carboplatin, number of wTP cycles, and presence and severity of hypersensitivity reaction to cisplatin. The primary endpoint was the incidence of patients who were able to receive wTP for at least 1 cycle throughout the study period. The secondary endpoint was treatment efficacy. Patients who completed wTP were defined as follows: (1) received more than six cycles of wTP without a hypersensitivity reaction or disease progression, (2) experienced a hypersensitivity reaction to cisplatin but the hypersensitivity reaction occurred after six cycles, and (3) received a total of six cycles of platinum chemotherapy together with the previous regimen (carboplatin). We also investigated the response rate of wTP, progression-free survival (PFS), and overall survival (OS).

Tumor response was evaluated based on the Response Evaluation Criteria in Solid Tumors, version 1.1. A complete response (CR) was defined as disappearance of all assessable target lesions without evidence of new lesions. Partial response (PR) was defined as at least 30% reduction in the sum of the longest diameter of all target lesions. Progressive disease (PD) was defined as at least a 20% increase in the sum of the longest diameter of all target lesions or development of new lesions. Stable disease (SD) was defined as any condition not meeting the aforementioned criteria.

### 2.2. Protocol for wTP

Similar to previous reports [17,18], patients received paclitaxel (80 mg/m^2^) and cisplatin (25 mg/m^2^) on days 1, 8, and 15 of a 4-week period. If the patient’s condition was good, the 1-week washout was omitted. Paclitaxel and cisplatin were dissolved separately in 250 cc of saline, and paclitaxel was administered over 1 h, followed by cisplatin over 1 h. We instructed patients to drink 1 L of oral rehydration solution prior to treatment and administered no further infusion. Dexamethasone (9.9 mg), d-chlorpheniramine maleate (5 mg), and the histamine H2-receptor antagonist famotidine (20 mg) were administered as premedication on the day. Full premedication on the previous day was not performed.

### 2.3. Severity of Hypersensitivity Reaction

The severity of the hypersensitivity reaction was graded according to the allergic reaction category of the Common Terminology Criteria for Adverse Events (CTCAE) ver. 4.0: Grade 1 (G1): transient flushing or rash, drug fever < 38 °C, intervention not indicated, Grade 2 (G2): intervention or infusion interruption indicated, responds promptly to symptomatic treatment, prophylactic medication indicated for ≤24 h, Grade 3: prolonged, recurrence of symptoms following initial improvement, hospitalization indicated for clinical sequelae, Grade 4: life-threatening consequences, urgent intervention indicated, and Grade 5: death. The hypersensitivity reaction to carboplatin was confirmed when allergic reactions occurred during or at the end of carboplatin infusion.

### 2.4. Statistical Analysis

Pearson’s chi-test was used for categorical variables. The Kaplan–Meier method was used to estimate the cumulative incidence of hypersensitivity reaction, the median PFS, and OS curves for each treatment arm. Comparisons between the intergroup were performed using two-sided log-rank and Wilcoxon square tests. PFS was defined as the interval between the administration of cisplatin and progressive disease or death. OS was defined as the interval between cisplatin administration and death. All data were analyzed based on the intention-to-treat principle. All tests were two-sided. A *p*-value of <0.05 was considered statistically significant. All analyses were performed using JMP software, version 11.0 (SAS, Cary, NC, USA).

## 3. Results

### 3.1. Patients Characteristics

Patient characteristics are shown in Table 1. The median age of the patients was 59 years. The primary site was the ovary in 53 (62%) patients, the fallopian tube in 26 (30%) patients, and the peritoneum in 7 (8%) patients. Of the 86 patients, 71 (83%) had advanced disease (stage III/IV) at the initial diagnosis. High-grade serous carcinoma was observed in 60 (70%) patients, followed by clear cell carcinoma in 16 (19%) patients. A total of 73 patients (85%) developed a hypersensitivity reaction to carboplatin when receiving weekly paclitaxel and carboplatin ± bevacizumab in the previous platinum-based chemotherapy. The median number of carboplatin cycles administered before the occurrence of hypersensitivity reaction was 8 (IQR: 6–11, range: 1–24). The hypersensitivity reactions to carboplatin were graded as G1, G2, and G3 in 57 (66%), 26 (30%), and 1 (1%) patient, respectively. The number of patients who received wTP during the first, second, and third or more lines was 21 (24%), 35 (41%), and 30 (35%) patients, respectively. A total of 35 patients (41%) received wTP with bevacizumab.

### 3.2. Safety

WTP was performed for a median of four cycles (IQR: 3–7), and 71 patients (83%) were able to administer cisplatin for at least one cycle. Hypersensitivity reaction to cisplatin was observed in 15 patients (10 at G1 and 5 at G2). There were no deaths in all cohorts. The cumulative rate of hypersensitivity reaction to cisplatin was 17% (Figure 1). Seven (47%) and eight (53%) patients experienced hypersensitivity reactions to cisplatin within fewer than four and after five or more cycles of wTP, respectively. Two patients experienced hypersensitivity to cisplatin during the first cycle of wTP. In all 15 patients, the hypersensitivity reaction was ameliorated by steroid administration. There was no association between the occurrence of hypersensitivity reaction to cisplatin and the addition of bevacizumab (no bevacizumab: 6/51; 11.8% vs. bevacizumab: 9/35; 25.7%, *p* = 0.094).

### 3.3. Relationship between the Severity of Carboplatin Hypersensitivity and the Occurrence of Cisplatin Hypersensitivity

The incidence of hypersensitivity reaction to cisplatin was 11/58 (19%) in patients who experienced a G1 hypersensitivity reaction to carboplatin and 3/27 (11%) in patients who experienced a G2 hypersensitivity reaction to carboplatin. One patient experienced unknown grade of hypersensitivity reaction to carboplatin. There was no association between the severity of the hypersensitivity reaction to carboplatin and the occurrence of hypersensitivity reaction to cisplatin (*p* = 0.363).

### 3.4. Efficacy of Treatment

A total of 55 patients (64%) completed wTP treatment, as defined in the Methods Section: 9 (10%) patients discontinued due to hypersensitivity reaction to cisplatin within 6 cycles, 1 (1%) patient discontinued due to renal toxicity (grade 2) at the 6th cycle, and 21 (24%) patients discontinued due to progressive disease within 6 cycles (Figure 2).

The response of 61 patients who had measurable disease was as follows: CR 14 (23%), PR 16 (26%), SD 13 (21%), and PD in 18 patients (30%). Notably, the response rate was 49%. We could not evaluate the response of 25 patients who received one or two cycles of wTP after carboplatin hypersensitivity reaction or who received wTP during adjuvant chemotherapy after debulking surgery.

The median PFS and OS after the initiation of wTP was 10.6 months (95% CI: 7.7–17.7) and 25.9 months (95% CI: 19.0–50.2), respectively (Figure 3). For patients who developed a hypersensitivity reaction to carboplatin during first-line chemotherapy (*n* = 21, stage III: 10 patients, stage IV: 11 patients), the median PFS was 24.4 months (95% CI: 10.6–44.6) and OS was 50.2 months (95% CI: 14.0–not reached), respectively (Figure 4).

To evaluate the treatment efficacy of wTP during initial treatment, we have compared the prognosis between 21 patients who received wTP with carboplatin hypersensitivity reaction and 334 patients who received weekly paclitaxel and carboplatin without carboplatin hypersensitivity reaction in the initial treatment at our institution. We show the patient characteristics in Table A1 and survival analyses in Figure A1 of the Appendix. There was no significant difference between the intergroups in PFS and OS. The treatment efficacy of wTP was not inferior to those of weekly paclitaxel and carboplatin in the initial treatment.

## 4. Discussion

### 4.1. Key Findings of This Study

This study investigated consecutive patients over an 8-year period who received identical regiments of wTP owing to the occurrence of hypersensitivity reaction to carboplatin. The safety and efficacy of cisplatin administration was confirmed in patients with hypersensitivity reactions to carboplatin. Among the studies reported to date, it is the most uniform and has the highest number of patients. In total, 86 patients who developed hypersensitivity reactions to carboplatin were treated with wTP, and 83% of the patients were able to continue the treatment without a hypersensitivity reaction to cisplatin There were no deaths due to hypersensitivity reaction to cisplatin. As shown in this study and reports to date, it is acceptable to administer cisplatin to patients who developed hypersensitivity reactions to carboplatin, and wTP can provide a clinical biomarker to show that patients receiving this treatment are platinum-sensitive [8]. This treatment strategy may serve as a bridge to PARP inhibitor maintenance therapy.

### 4.2. Safety of wTP

83% of the patients received treatment without a hypersensitivity reaction to cisplatin, and there were no deaths due to the hypersensitivity reaction to cisplatin. In a previous report, Bergamini et al. reported that 5 of 38 patients (13.2%) developed a hypersensitivity reaction to mild or moderate cisplatin [15]. Kolomeyevskaya et al. analyzed 19 patients who received cisplatin but observed no hypersensitivity reaction [19]. Callahan et al. also reported that 11 of 59 patients (18.6%) who developed hypersensitivity reaction to carboplatin [14], including 24 patients in their experience and 35 patients in previous reports, experienced hypersensitivity reaction after the administration of cisplatin; of these 59 patients, 2 died [20,21]. Combining their data [14,15,19] with our report, the incidence of the hypersensitivity reaction to cisplatin in patients with hypersensitivity reaction to carboplatin was 31/202 (15.3%), with two deaths (1.0%). In addition, there was no association between the severity of the hypersensitivity reaction to carboplatin and the incidence of hypersensitivity reaction to cisplatin in our study. Although both carboplatin and cisplatin are platinum agents, their cross-reactivity is reported to be low [22].

### 4.3. Efficacy of wTP

In terms of long-term efficacy, the majority of patients (54 patients, 63%) completed the scheduled wTP, including 6 patients who experienced hypersensitivity reaction to cisplatin after 6 cycles. Only 9 patients (10%) discontinued treatment within 6 cycles due to hypersensitivity reaction to cisplatin. In terms of the use of PARP inhibitors [8], patients who received more than 4 cycles of wTP and had complete or partial responses would be able to receive the benefits of a PARP inhibitor.

### 4.4. Weekly versus Triweekly Carboplatin in Hypersensitivity Reaction

Compared with previous studies, a large sample size was obtained despite this being a single-institution study. One of the reasons may be that we chose weekly paclitaxel-carboplatin chemotherapy as the first-line chemotherapy, which resulted in a higher incidence of hypersensitivity reaction to carboplatin than triweekly administration. Reportedly, the incidence of hypersensitivity reactions to carboplatin is higher in weekly treatment for low-grade glioma in children [23,24]. A higher incidence of hypersensitivity reaction has been reported in weekly paclitaxel–carboplatin chemotherapy when used as a primary treatment for advanced ovarian cancer [25,26]. The International Collaboration on Ovarian Neoplasms (ICON)-8 study [27] revealed that allergic reactions (of any grade) were significantly higher in the group of patients administered weekly carboplatin compared with triweekly carboplatin (triweekly carboplatin: 110/911 patients, vs. weekly carboplatin: 90/420 patients; *p* < 0.001).

### 4.5. Advantages of Weekly Paclitaxel and Cisplatin Chemotherapy

We chose weekly administration of paclitaxel and cisplatin because the use of weekly cisplatin reduces gastrointestinal symptoms, such as nausea, vomiting, and renal toxicity. A previous phase III study reported that gastrointestinal symptoms and renal toxicity were higher in triweekly cisplatin than triweekly carboplatin [28,29]. Moreover, dose-dependent administration of paclitaxel has been shown to be effective in Japan [30]. Reportedly, this regimen has mild renal toxicity and hematologic toxicity, even in other cancers [17,18]. Our historical cohort analysis revealed that the treatment efficacy of wTP was not inferior to those of weekly paclitaxel and carboplatin in the initial treatment. The administration of cisplatin may be useful in terms of safety and efficacy for patients with hypersensitivity reactions to carboplatin in first-line chemotherapy treatment.

### 4.6. Strengths and Limitation

The strengths of this study include a large and homogenous cohort of all 86 patients who received wTP for a platinum-sensitive disease. The limitation was that the use of a weekly schedule of platinum is uncommon, at least in Europe and the USA, along with the retrospective design. 

## 5. Conclusions

The administration of cisplatin to patients with hypersensitivity reaction to carboplatin was a concern owing to two deaths reported previously [20,21]. However, when combining the results of our study and previous reports [14,15,19], the incidence of hypersensitivity reaction to cisplatin was shown to be 31/183 (16.9%), with two deaths [16,17] (1.1%). While desensitization therapy is cumbersome and there are disadvantages in terms of temporal constraints, wTP is simple and easy to administer. It is acceptable to administer cisplatin to patients who develop a hypersensitivity reaction to carboplatin. Although the use of a weekly schedule of platinum is uncommon, at least in Europe and the USA, this treatment strategy may serve as a bridge to PARP inhibitor maintenance therapy.

## Figures and Tables

**Figure 1 cancers-13-00640-f001:**
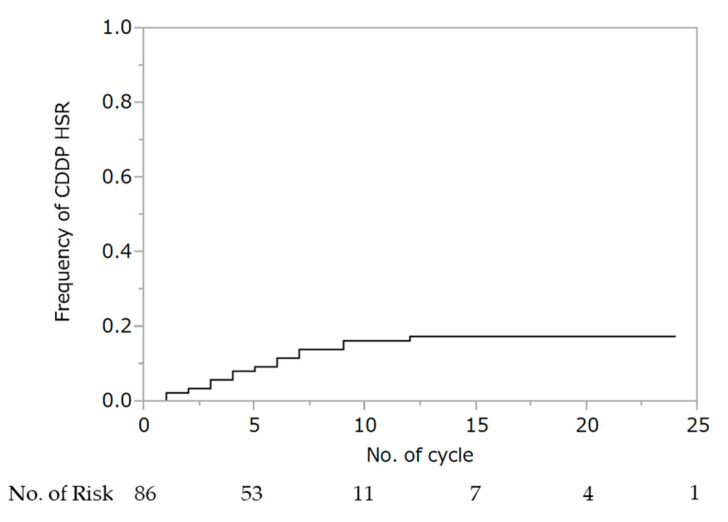
Cumulative rate of hypersensitivity reaction to cisplatin. Abbreviations: HSR, hypersensitivity reaction.

**Figure 2 cancers-13-00640-f002:**
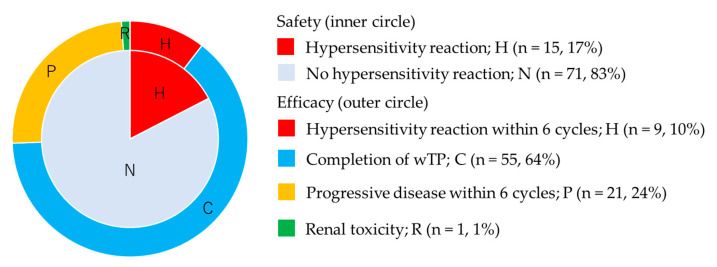
Safety and long-term efficacy of weekly paclitaxel and cisplatin. Inner circle shows safety and outer circle shows long-term efficacy.

**Figure 3 cancers-13-00640-f003:**
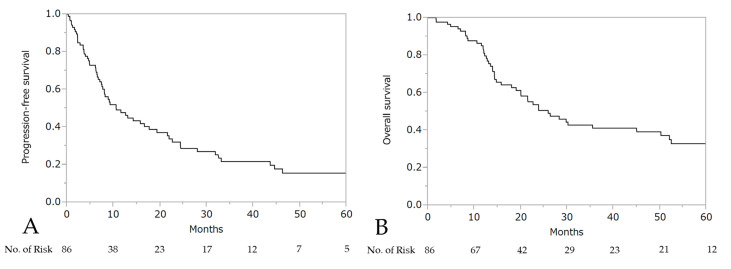
Progression-free survival (PFS) (**A**) and overall survival (OS) (**B**) after the initiation of wTP.

**Figure 4 cancers-13-00640-f004:**
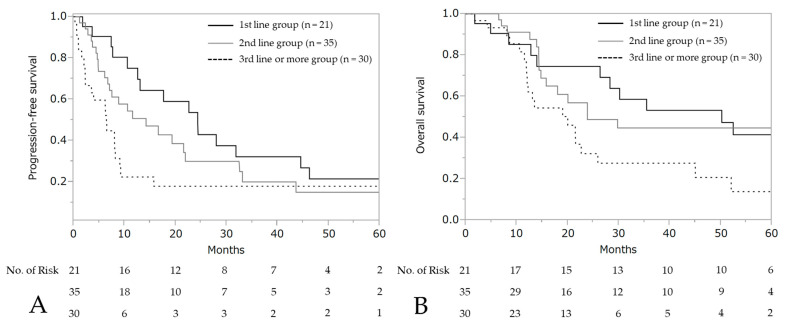
Progression-free survival and overall survival according to each line. Progression-free survival (**A**) and overall survival (**B**) after wTP initiation.

**Table 1 cancers-13-00640-t001:** Patients characteristics.

Variations	No. of Patients	Incidence (%)
Age (years)		
Median, (IQR)	59 (51–67)	
Primary site		
Ovary	53	62
Peritoneum	7	8
Fallopian Tube	26	30
Primary FIGO stage		
I/II	15	17
III/IV	71	83
BRCA status		
BRCA 1 mutation+	5	6
BRCA 2 mutation+	1	1
BRCA negative	11	13
Unknown	69	80
Histology		
Serous carcioma, high-grade	60	70
Endometrioid carcinoma	4	5
Clear cell carcioma	16	19
Others	6	7
Previous platinum-based chemotherapy		
Triweekly paclitaxel (docetaxel) and carboplatin ±Bev	13	15
Weekly paclitaxel and carboplatin ±Bev	73	85
No. of cycles in previous carboplatin		
Median, (IQR)	8 (6–11)	
2–6	28	33
7–12	42	49
13–	16	19
Grade of previous carboplatin hypersensitivity reaction		
1	57	66
2	26	30
3	1	1
unknown	1	1
Timing of weekly paclitaxel and cisplatin		
Initial treatment	21	24
First recurrent treatment	35	41
Second or more recurrent treatment	30	35

Abbreviations: Bev, bevacizumab; BRCA; breast cancer susceptibility, FIGO, International Federation of Gynecology and Obstetrics; IQR, interquartile range.

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
