# Peer review of "Safety and Efficacy of Weekly Paclitaxel and Cisplatin Chemotherapy for Ovarian Cancer Patients with Hypersensitivity to Carboplatin"

_cancers, 2021, doi:10.3390/cancers13040640_

Round 1

Reviewer 1 Report

Authors should double check all the references 

Title: I suggest to add the adjective “recurrent” ovarian cancer

Abstract: I suggest to specify the retrospective nature of the study in the Methods session, for example adding: “We retrospectively investigated”

Lines 27 and 29: what is the “efficacy for treatment”? PFS and OS are the endpoint for efficacy. I think that Authors should change in “treatments compliance”

Line 34: the conclusion should be “wTP was safe and well tolerated in patients who developed carboplatin HSR.  

Lines 40-41: I disagree with this statement. The standard chemotherapy for platinum sensitive recurrent OC is a platinum based chemotherapy plus BEV or PARPi maintenance in case of response to platinum and it is now influenced by the first line treatment (use of Bev or PARPi).  The choice of the best targeted therapy to add to platinum has been investigated in this network meta-analysis: Bartoletti M, Pelizzari G, Gerratana L, Bortot L, Lombardi D, Nicoloso M, Scalone S, Giorda G, Baldassarre G, Sorio R, Puglisi F. Bevacizumab or PARP-Inhibitors Maintenance Therapy for Platinum-Sensitive Recurrent Ovarian Cancer: A Network Meta-Analysis. Int J Mol Sci. 2020 May 27;21(11):3805. doi: 10.3390/ijms21113805. PMID: 32471250; PMCID: PMC7312982

Moreover, we now know that the best chemo backbone to add to BEV is carbo + PLD (Pfisterer, J. et al. (2020) ‘Bevacizumab and platinum-based combinations for recurrent ovarian cancer: a randomised, open-label, phase 3 trial’, The Lancet Oncology, 21(5), pp. 699–709. doi: 10.1016/S1470-2045(20)30142-X.).

The first reference is about 18 years old. I suggest to change it. The second reference is referring to the impact of secondary cytoreduction that is very important in platinum sensitive recurrent OC but is not cite in the sentence. 

Line 73: the statement “prospectively enrolled database of patients ” used in this context is misleading, I suggest to change “ A series of 86 consecutive patients was retrospectively enrolled  among those receiving chemotherapy at our institute  from January 2011 to December 2019 .

Lines 77-78 “and wTP was performed at the chemotherapy unit for outpatients. “ This sentence is not useful and can be deleted. 

Lines 78-79. I think that Authors have confused words carboplatin and cisplatin 

Lines 78-80: It is not usual to call the treatment at disease recurrent as First line or second lines. I suggest to chance to “treatment at first recurrence” and “treatment at second disease recurrence”. Table 1 should be modify accordingly (Timing of wTP administration: 1) first disease recurrence ) second disease recurrence….)

Lines 81-83: Authors should specified the exact number of patients who were treated with carbo at a slow infusion rate following HSR

Lines 221-221: Authors should check the references as they are incorrect and referred to NACT. Moreover, the Authors should recognize that the positive impact of a weekly schedule has been demonstrated for Asian patient only (JGOG3016 trial ). For caucasian patients MITO 7 and ICON 8 have failed to demonstrate a superiority of the weekly schedule

Line 234: in this paragraph authors should recognized that the use of weekly schedule of platinum is uncommon in the recurrent setting, at least in Europe and USA. Thus, it represent a major limitation of this work along with the retrospective design. 

Lines 240-248: In these lines there is a repetition of data yet reported in results and discussion. Authors can save this words to add the Reviewers’ comments

Lines 253-254: I can’t understand the meaning of this sentence “and wTP provides a clinical biomarker to show that patients treated with this treatment are platinum-sensitive [5]. “ I think that this statement should be omitted leaving the concept expressed in line 255

Reviewer 2 Report

in the attachment

Reviewer 3 Report

Tate S. et al evaluated the safety and efficacy of weekly paclitaxel and cisplatin chemotherapy in patients with who developed hypersensitivity reaction to carboplatin. Despite being a single-institution study, the authors included 86 patients with ovarian, fallopian tube, and peritoneal carcinoma who developed carboplatin HSR during previous chemotherapy (carboplatin+paclitaxel).

The data analysis and the interpretation of the results are robust.  The results show that 83% of the patients were able to receive treatment without HSR to cisplatin. HSR to chemotherapy drugs is an important aspect in oncology treatments as HSR could lead to treatment interruption leaving patients with limited treatment options. Therefore, is important to find and test alternatives that are effective without (or minimum) HSR. This report adds more information about HSR to carboplatin, is well written and has a scientific soundness.

 Minor alterations:

  • Figure 3 and 4 are incomplete (right side)
  • The conclusion could be more robust e assertive.

Round 2

Reviewer 1 Report

The manuscript has been successfully improved after the Author’s revisions.  I’ve appreciated the Authors response to my observations. Authors should keep an eye on line 263 (maybe the right word is historical not histological).

Author Response

Reviewer 1:

Comments(1)

The manuscript has been successfully improved after the Author’s revisions.  I’ve appreciated the Authors response to my observations. Authors should keep an eye on line 263 (maybe the right word is historical not histological).

Response

Thank you for pointing it out. We changed to “historical”.

Reviewer 2 Report

Introduction

Page 1 line 41. The standard ‘chemotherapy’ for …..

Do authors indicate 'therapy' since you provide information with combination treatments more than chemotherapy alone?

Page 2 lines 45-46

English wording should be improved, such as 'the best targeted therapy of choice' 'added to' platinum....; 'the best chemotherapy  backbone' 'combined with Bev' ....

Please introduce what is the main results of the network meta-analysis. That will be helpful information for readers.

Statistical analysis

Page 4 line 135  'Pearson's chi-square' …

Page 4 line 136  The wording should be improved, such as 'the cumulative incidence of ....'

Results

3.4. Efficacy of treatment

Page 7 line 180 Generally, this article should be improved with the assistance of a native English speaker. (incorrect spelling, inconsistent abbreviations, eg. wTP and WTP, etc...)

Discussion

Page 10 line 206

4.1. Safety and efficacy of wTP

In this paragraph, authors may condense your most important findings but not repeated results already shown in Results section (eg. Lines 210-214)

Its subtitle may not be necessary since authors has two similar subtitles for discussion hereafter (subtitle 4.2 and 4.3).

Page 10 line 244

4.4 Incidence of hypersensitivity reaction in weekly carboplatin chemotherapy

The subtitle may be improved to fulfill authors point, eg. Weekly versus triweekly carboplatin in hypersensitivity reaction

Page 10 line 245

This sentence could be moved to Strength Section. Hence, the following sentence should be rewritten, eg. 'Compared with the literature[??], a higher incidence of hypersensitivity reaction to carboplatin in our study may be resulted from the included patients, mostly treated with weekly paclitaxel-carboplatin as front-line therapy.   

Page 10 line 256

4.5 Weekly paclitaxel and cisplatin chemotherapy

The subtitle may be improved to fulfill authors point, eg. the advantages and disadvantages of wTP regimen.

Author Response

Reviewer 2:

Comments(1)

Page 1 line 41. The standard ‘chemotherapy’ for …..

Do authors indicate 'therapy' since you provide information with combination treatments more than chemotherapy alone?

Response

We changed “chemotherapy” to “therapy”.

Comments(2)

Page 2 lines 45-46

English wording should be improved, such as 'the best targeted therapy of choice' 'added to' platinum....; 'the best chemotherapy  backbone' 'combined with Bev' ....

Please introduce what is the main results of the network meta-analysis. That will be helpful information for readers.

Response

We changed to the following sentences: PARP inhibitors have been reported to be more effective in survival than Bev in patients with BRCA mutations in a network meta-analysis. Moreover, a recent study has reported that carboplatin plus pegylated liposomal doxorubicin is the best chemotherapy regimen in combination with Bev for platinum-sensitive recurrent ovarian cancer.

Comments(3)

Statistical analysis

Page 4 line 135  'Pearson's chi-square' …

Page 4 line 136  The wording should be improved, such as 'the cumulative incidence of ....'

Response

We changed to “Pearson’s'”.

We also changed to the sentence “The Kaplan-Meier method was used to estimate the cumulative incidence of hypersensitivity reaction, the median PFS, and OS curves for each treatment arm.”

Comments(4)

Results

3.4. Efficacy of treatment

Page 7 line 180 Generally, this article should be improved with the assistance of a native English speaker. (incorrect spelling, inconsistent abbreviations, eg. wTP and WTP, etc...)

Response

This manuscript has been edited by an English editing service. We carefully checked spelling and abbreviations again. We change the sentences in this paragraph.

Comments(5)

Discussion

Page 10 line 206

4.1. Safety and efficacy of wTP

In this paragraph, authors may condense your most important findings but not repeated results already shown in Results section (eg. Lines 210-214)

Its subtitle may not be necessary since authors has two similar subtitles for discussion hereafter (subtitle 4.2 and 4.3).

Response

We changed the subtitle to “Key findings of this study”. We deleted the repeated sentences in this paragraph.

Comments(6)

Page 10 line 244

4.4 Incidence of hypersensitivity reaction in weekly carboplatin chemotherapy

The subtitle may be improved to fulfill authors point, eg. Weekly versus triweekly carboplatin in hypersensitivity reaction

Response

We changed the subtitle to “Weekly versus triweekly carboplatin in hypersensitivity reaction”.

Comments(7)

Page 10 line 245

This sentence could be moved to Strength Section. Hence, the following sentence should be rewritten, eg. 'Compared with the literature[??], a higher incidence of hypersensitivity reaction to carboplatin in our study may be resulted from the included patients, mostly treated with weekly paclitaxel-carboplatin as front-line therapy.   

Response

Thank you for your suggestion. We already described in the strength and limitation section that “The strengths of this study include a large and homogenous cohort of all 86 patients who received wTP for a platinum sensitive disease”. Therefore, we did not move the sentences.

In the 4.4 pargraph in discussion section, we would like to claim that a large sample size was obtained because most patients included in this study received weekly paclitaxel-carboplatin chemotherapy as the first-line chemotherapy. We described the comparision between weekly and triweekly administration in hypersensitivity reaction to carboplatin in this paragraph. Therefore, we changed the subtitle to “Weekly versus triweekly carboplatin in hypersensitivity reaction” as reviewer’s suggestion. We did not show the incidence of hypersensitivity reaction to carboplatin in this study, but we included the large number of patients with hypersensitivity reaction to carboplatin in this study .

Comments(8)

Page 10 line 256

4.5 Weekly paclitaxel and cisplatin chemotherapy

The subtitle may be improved to fulfill authors point, eg. the advantages and disadvantages of wTP regimen.

Response

Thank you for your suggestion.

We changed the subtitle to ”Advantages of weekly paclitaxel and cisplatin chemotherapy” because this paragraph did not describe the disadvantage.